# Advancements in Characterizing *Tenacibaculum* Infections in Canada

**DOI:** 10.3390/pathogens9121029

**Published:** 2020-12-08

**Authors:** Joseph P. Nowlan, John S. Lumsden, Spencer Russell

**Affiliations:** 1Department of Pathobiology, University of Guelph, Guelph, OT N1G 2W1, Canada; jsl@uoguelph.ca; 2Center for Innovation in Fish Health, Vancouver Island University, Nanaimo, BC V9R 5S5, Canada; spencer.russell@viu.ca

**Keywords:** *Tenacibaculum*, tenacibaculosis, fishes, bivalves, aquaculture

## Abstract

*Tenacibaculum* is a genus of gram negative, marine, filamentous bacteria, associated with the presence of disease (tenacibaculosis) at aquaculture sites worldwide; however, infections induced by this genus are poorly characterized. Documents regarding the genus *Tenacibaculum* and close relatives were compiled for a literature review, concentrating on ecology, identification, and impacts of potentially pathogenic species, with a focus on Atlantic salmon in Canada. *Tenacibaculum* species likely have a cosmopolitan distribution, but local distributions around aquaculture sites are unknown. Eight species of *Tenacibaculum* are currently believed to be related to numerous mortality events of fishes and few mortality events in bivalves. The clinical signs in fishes often include epidermal ulcers, atypical behaviors, and mortality. Clinical signs in bivalves often include gross ulcers and discoloration of tissues. The observed disease may differ based on the host, isolate, transmission route, and local environmental conditions. Species-specific identification techniques are limited; high sequence similarities using conventional genes (16S rDNA) indicate that new genes should be investigated. Annotating full genomes, next-generation sequencing, multilocus sequence analysis/typing (MLSA/MLST), matrix-assisted laser desorption/ionization time-of-flight mass spectrometry (MALDI-TOF), and fatty acid methylesters (FAME) profiles could be further explored for identification purposes. However, each aforementioned technique has disadvantages. Since tenacibaculosis has been observed world-wide in fishes and other eukaryotes, and the disease has substantial economic impacts, continued research is needed.

## 1. Introduction

Salmonid aquaculture started in Canada roughly 200 years ago, and as of 2017, Canada is now the fourth-largest supplier of salmonid products [1,2,3]. Recently, salmon farmers in British Columbia (BC; Canada) have experienced frequent mortality events in Atlantic salmon (*Salmo salar*, AS) due to ‘mouthrot’ (a unique form of tenacibaculosis (other common names synonymous with mouthrot include yellow-mouth and bacterial stomatitis)) [4,5,6,7,8]. Mortality events in farmed fishes due to tenacibaculosis have also been experienced worldwide. Current research has shown that multiple species within the genus *Tenacibaculum* could be an etiological agent responsible for tenacibaculosis [6,7,9,10,11,12,13,14,15,16].

The genus *Tenacibaculum* is critically understudied. Significant knowledge gaps exist regarding bacterial diversity, distribution, and pathogenicity of *Tenacibaculum* species. Given that the economic impact of mouthrot seems to be increasing, it is imperative that research focuses on the pathogenesis of disease and the role of the various bacterial species that induce mouthrot. Therefore, the principal objective of this publication is to review the ecology, identification, and impacts of potentially pathogenic species, with a focus in Canada if possible. Garnered information will allow interpretations of the advancements made in characterizing *Tenacibaculum* sp. infections.

## 2. Literature Review

### 2.1. Economics, Treatments and Environmental Impacts

From 2016 to 2018, ~122,420 tons of salmon was produced in Canada yearly (72% from BC) that was annually valued at ~$1 billion [17]. No country-wide or global report on the damage caused by *Tenacibaculum* spp. has been developed; however, workshops describing global issues associated with the genus have occurred [4,5]. Within Canada, representatives of Grieg Seafoods (GS) described that in 2014, *Tenacibaculum* outbreaks in AS cost over $1.6 million, attributed to treatments and reduced annual profits from a loss in growth [5]. Four years later, a veterinarian from a local AS aquaculture company described that a single outbreak cycle at a single AS site might cost up to $500 K [18]. Furthermore, outbreaks by *Tenacibaculum* species are a global issue and have major commercial impacts on salmonid and other species production.

At three commercial AS netpen sites in BC from 2015–2016, mouthrot outbreaks were generally characterized by a sharp increase in mortalities with clinical signs of mouthrot compared to the baseline. Subsequent spikes in fish mortality with clinical signs of mouthrot ranged from hundreds to thousands per day before or during the application of antimicrobials. By the end of treatment, a reduction in mortalities to baseline levels typically occurred and rapid application of antimicrobials were key to reduce daily mortality. However, at numerous sites, repeated outbreaks are observed following the introduction of naïve post-smolts. Repeated outbreaks have also been reported in the same netpen after treatment. Important challenges to successful management include the length of time between diagnosis and delivery of antimicrobials and that affected fish tend to have reduced feeding rates. There are also numerous unknowns that impede a description of an outbreak cycle at netpen sites, including the limitations of diagnosis and various environmental, host, and pathogen factors. Diagnosis is predominantly based on clinical signs in dead fish and knowledge of the causative species of bacteria and is lacking or controversial. The impact of numerous environmental, host, and pathogen factors likely lead to discrepancies in the magnitude of mortalities between sites, however these are largely undescribed.

In Canada, only a few antimicrobial agents are available to treat food fish including florfenicol (Aquaflor) to trimethoprim and sulfadiazine (Romet 30) [19], both of which are bacteriostatic. To treat tenacibaculosis, local AS aquaculture companies use antimicrobials such as Romet 30 or Aquaflor [5,18,19,20]. Aquaflor is widely applied because fish can be harvested 12 days after treatment, in comparison to Romet 30, where fish can be harvested 42 days after treatment [19]. Since naïve post-smolts are not at harvestable sizes during mouthrot outbreaks and are the predominant group of AS to display mouthrot, a justification for using florfenicol can be based on the effective dose to treat fish, which limits antimicrobial use. The recommended dose of Aquaflor is 10 mg kg^−1^ [21], in contrast to Romet 30, which is 167 mg kg^−1^ [22]. Continuous use of single or few antimicrobials to treat tenacibaculosis may lead to resistance. In addition, florfenicol is expensive and repeated treatments increase productions costs. The number of antibiotic applications required to treat tenacibaculosis vary; however, the use is concentrated after smolt entry and typically only occurs after a sufficiently large spike in mortality [5].

Pathogenic effects induced by *Tenacibaculum* spp. are predominantly recorded in fishes and are most often identified in aquaculture or laboratory settings, but disease is expected to occur in wild populations. However, investigations of tenacibaculosis in wild populations are limited; one study identified ulcers on white seabass (*Atractoscion nobilis*) sampled at Redondo and Newport Beach (California, USA), which were comparable to the ulcers induced by *T. maritimum* [23]. In another study, Atlantic cod (*Gadus morhua*) sampled in the Wadden Sea (Germany) displayed ulcers and yellow plaques around the mouth that were believed to be caused by *T. maritimum* [24]. *Tenacibaculum* infections may have also been identified in wild Picasso triggerfish, black damselfish, striped trumpeter, and turbot [25,26,27], but cannot be confirmed because fishes were caught and housed in aquaria before the diagnosis of disease. It will be important to identify more cases of disease in wild populations, to better understand the impact that *Tenacibaculum* spp. have on the environment.

### 2.2. Tenacibaculum Biology

#### 2.2.1. Family Introduction

The family *Flavobacteriaceae* potentially encompasses over 90 genera and hundreds of species [28]. The genus *Flavobacterium*, the first identified genus in *Flavobacteriaceae*, originally contained 26 species of bacteria that were identified using dichotomous characteristics [29], including the inability to degrade agar, alginates, or chitin, the inability to produce recordable amounts of acidic compounds when given certain sugars, and the development of non-water soluble yellow, orange, brown, or red pigments on various media [29]. Variable characteristics between the 26 species included that bacteria were Gram-negative or Gram-positive, composed of short rods to filamentous fibers, and were either motile through peritrichous flagella, or non-motile [29]. By 1978, the genus became a catch-all for bacteria that did not fit easily into other taxa [28,30]. The phylogenetic placement of the genus was shifted and was eventually placed in the family *Bacteriaceae* based on dichotomous characteristics and guanine-cytosine (GC) contents [30].

By 1996, it was recognized that 16S rDNA oligonucleotide sequences, DNA-rRNA hybridization data, and GC contents indicated that the genera *Flavobacterium*, *Cytophaga*, and *Flexibacter* are highly polyphyletic, and further indicated that bacterial groups might have misplaced taxonomic positions [31]. Using morphological characteristics and 23S and 16S rDNA similarity dendrograms from melting temperature, it was suggested that only 10 species could be classified as *Flavobacterium* [31]. The family *Flavobacteriaceae* was originally proposed using only morphological characteristics [32]; however, the emended description was based on both genetic and morphological criteria. The family includes bacteria, which are Gram-negative, display short rods to filamentous fibers, can be motile or non-motile, are pigmented or non-pigmented, are chemoorganotrophic, have no sphingophospholipids, have menaquinone-6 as the major respiratory quinone, cannot degrade cellulose, are often saprophytic, can be aquatic or terrestrial [28,31], and have a GC content ranging from 29% [31] to 55% [28]. Later, the minimal standards for describing new taxa of *Flavobacteriaceae* were developed based on bacterial morphology, DNA hybridization, and 16S rDNA sequences [33]. The present standard to determine the species and strain level of several families, including *Flavobacteriaceae*, are 16S rDNA sequences [28,32,34]. However, *Flavobacterium* and *Tenacibaculum* sometimes require other housekeeping genes such as gyrase subunit B (gyrB) to determine species and strain taxa, as some species can have high (>99%) sequence similarities using conventional genes [35,36].

Members of the family *Flavobacteriaceae* tend to be ubiquitous in terrestrial and aquatic environments. Most known members of this family have mutualistic relations with host organisms or have been found in environmental samples [28]. Some species within the family such as *Flavobacterium psychrophilum*, *Flavobacterium columnare*, *Tenacibaculum maritimum*, *Riemerella anatipestifer*, *Ornithobacterium rhinotracheale*, *Coenonia anatine*, *Capnocytophaga canimorsus*, and *Elizabethkingia meningoseptica* are pathogenic to select groups of animals [28]. *F. psychrophilum* is an etiological agent of bacterial cold-water disease (BCWD) and rainbow trout fry syndrome (RTFS) [37,38], while *F. columnare* is an etiological agent of columnaris disease [39]. Both *T. maritimum* [6,7] and *T. finnmarkense* [11,40] are thought to be etiological agents of tenacibaculosis in fishes. There are some similarities between columnaris disease [39] or BCWD/RTFS [37,38] with some clinical presentations of tenacibaculosis, where disease results in large ulcers on epidermal surfaces of fishes [6,7,9,10,11,12,13,14,15,16,40]. Similarities between the clinical presentations of columnaris disease, BCWD, RTFS, and tenacibaculosis may indicate that both genera of bacteria use similar strategies to induce disease in fishes.

#### 2.2.2. Genus Introduction

In the 21st century, *Tenacibaculum* species and strain identification are primarily based on 16S rDNA sequences and bacterial morphology. The genus *Tenacibaculum* was initially identified using 16S rDNA and gyrB sequences [41]. The phylogenetic analysis (neighbour-joining method) by Suzuki et al. (2001) using the partial gyrB gene sequence demonstrated that *Flexibacter* species were not closely related to *Tenacibaculum* species [41]. *Flexibacter flexilis* had gyrB sequence similarities between 69.4% and 76.4% compared to other *Flexibacter* species, including the reclassified *Flexibacter maritimus* (*Tenacibaculum maritimum*) and *Flexibacter ovolyticus* (*Tenacibaculum ovolyticum*) [41]. However, the sequence similarity comparing gyrB sequences of *T. ovolyticum* to *T. maritimum* was 95.2%. 16S rDNA sequences were similar among species tested and demonstrated a closely related phylogeny to the gyrB gene, where the phylogenetic sister to *Tenacibaculum* was proposed to be *Polaribacter* [41]. DNA-DNA hybridization also occurred; within a *Tenacibaculum* species, reassociation values were generally at or above 86 ± 2%, while outside the species and genus, reassociation values were less than 35 ± 6% and 18 ± 1%, respectively [41]. Overall, Suzuki et al. (2001) concluded that *Tenacibaculum* species were separate from *Flexibacter* and occupied a novel genus [41]. Other genes have also been applied for the identification of *Tenacibaculum* species, including *atpA*, *dnaK*, *glyA*, *gyrB*, *ileS*, *infB*, *rlmN*, *tgt*, *trpB*, *tuf*, and *yqfO* [35]. Phylogenies generated through multilocus sequence analysis (MLSA) and 16S rDNA were different [35]. If MLSA was used, there were potentially three monophyletic clades instead of several polyphyletic clades. Using MLSA, Clade 1 consisted of *T. mesophilum*, *T. aestuari*, *T. lutimaris*, *T. litoreum*, *T. discolor*, and *T. gallaicum*, Clade 2 consisted of *T. aiptasiae*, *T. ovolyticum*, *T. dicentrarchi*, and *T. soleae*, Clade 3 consisted of *T. geojense*, *T. skagerrakense*, *T. amylolyticum*, and *T. jejuense*, and there were independent lineages of *T. adriaticum* and *T. maritimum* together, and *T. litopenaei* and *T. crassotreae* separately [35]. MLSA is a more accurate technique compared to traditional phylogenies using only 16S rDNA because as more protein-encoding genes are compared, the resolution of phylogenetic analyses improves. However, the trade-off is the time allocated and the expenses applied. MLSA was also used with 7 of 11 housekeeping genes [35] to investigate the diversity of Norwegian *Tenacibaculum* isolates [42]. The MLSA phylogeny proposed by [42] differs from that proposed by [35]; differences include the relation and position of select species. These differences may be related to novel isolates used and different numbers (7 [42] vs. 11 [35]) of housekeeping genes used for comparisons. A recent review by Fernández-Álvarez et al. (2018) focused on *Tenacibaculum* species identification techniques using four methods: (1) culture-based, (2) serological studies and immunological, (3) genotyping and molecular, and (4) proteomic and chemotaxonomic [43].

A potential way to infer some aspects of *Tenacibaculum* spp. biology and one way to identify *Tenacibaculum* species are through complete genome sequence analyses. The full DNA sequence of *T. maritimum* NCIMB 2154^T^ has a 3,435,971 bp chromosome predicted to contain 3071 genes encoding 2866 proteins [44]. Complete genomes of other *Tenacibaculum* spp. have been shown to have similar characteristics (i.e., bp, GC%, number of genes and proteins) to *T. maritimum* NCIMB 2154^T^ (Table A1). The potential pathogenicity of *T. maritimum* NCIMB 2154^T^ has been predicted through complete genome sequencing and will be further discussed in Section 2.3.4 [44]. Full genome analyses would provide valuable information, and is slowly becoming cheaper; however, it is still too expensive to routinely sequence isolates collected from the field.

Bacteria have been traditionally identified by comparing physical morphology and biochemical characteristics; however, these comparisons alone do not always differentiate between species or genera. Most *Tenacibaculum* spp. are yellow; rod-shaped; of similar width (0.2–0.7 µm) and similar length (typically < 10 µm); lack flagella; and are capable of gliding motility (Table A2). They are also gram-negative, catalase-positive, oxidase-positive, strictly aerobic, and have similar pH tolerances (Table A2). In comparison, temperature and salinity tolerance and the ability to reduce nitrate are more variable characteristics among the identified species (Table A2). When describing bacterial morphology from culturing, one should note that variations could occur based on the media applied [43]. Medias applied to culture *Tenacibaculum* spp. include marine agar [9,16,43,45,46,47], *Flexibacter maritimus* medium [43,45,46,47], Anacker and Ordal agar with modifications [43,45,48,49], tryptone agar with modifications [43,47], thiosulphate-citrate-bile-sucrose agar [43,47], Marine Luria Broth medium prepared with seawater [41,43], *Cytophaga* agar with seawater [50], blood agar with modifications [16,25], and more; different chemical compositions are likely to be selective for particular isolates of *Tenacibaculum*. Selection for *Tenacibaculum* isolates can also occur by using media with aminoglycoside antibiotics such as kanamycin. The minimum inhibitory concentrations to kanamycin for *Tenacibaculum* are high (>30 µg per disc [51,52,53,54] up to 500 µg per disc [40] and 50 µg mL^−1^ [55]), relative to other bacterial groups around netpens [56]. Variations in bacterial characteristics can also change depending on when an isolate of *Tenacibaculum* is selected for sampling; prolonged periods in sub-cultured media can lead to the generation of spherical cells [53,57]. While there are multiple methods to identify and characterize *Tenacibaculum* spp. and strains, few molecular diagnostic techniques have been developed for this genus. Further research needs to focus on developing accurate and fast identification techniques for *Tenacibaculum* species.

#### 2.2.3. Distribution and Diversity

From the limited research performed to date, the genus *Tenacibaculum* appears to be very diverse and many more species are likely to be described. Thirty-two named species are currently described, predominatly through 16S rDNA sequencing (Table A2). Since 2006, 22 new species have been described.

The genus *Tenacibaculum* has a cosmopolitan distribution within saltwater; however, local distributions of *Tenacibaculum* spp. are largely unknown. Currently, 20 out of 32 species have only been identified in Asia, and five species are unique to Europe (Table A2). The restricted distribution of these species may be a result of the lack of investigation to date. The remaining seven species have broader distributions; *T. maritimum* has been found in marine waters in Canada (East and West coast), Chile, Japan, Norway, Ireland, Spain, and Australia; *T. dicentrarchi* has been identified in Antarctica, Canada (East and West coast), Chile, Norway, Spain, and potentially Australia; *T. finnmarkense* has been identified in Chile and Norway; *T. soleae* has been identified in Canada, the USA, Europe, and Australia; *T. xiamenense* has been identified in China and Chile; *T. mesophilum* has been identified in Japan, the USA, and China; and *T. ovolyticum* has been identified in Japan, the USA, and Norway (Table A2). Further research will likely reveal that many species have more cosmopolitan distributions.

#### 2.2.4. Host Relationships

Approximately three-quarters of known *Tenacibaculum* spp. have been found in only one or two hosts and have not been reported to be pathogenic (Table A2). Non-pathogenic *Tenacibaculum* spp. have been described from algae, tunicates, tidal sediments, seawater, mollusks, and crustaceans (Table A2). The remaining *Tenacibaculum* spp. have proposed pathogenic relationships with multiple fishes and few bivalves; these include *T. maritimum*, *T. dicentrarchi*, *T. finnmarkense*, *T. gallaicum*, *T. discolor*, *T. ovolyticum*, *T. mesophilum*, and *T. soleae* (Table A2). However, a pathogenic relationship fulfilling Koch’s postulates with a host species has only been demonstrated for *T. maritimum* [6] and *T. finnmarkense* [11,40] in *S. salar. T. maritimum* has also been reported in other animals such as sea lice (*Lepeophtheirus salmonis*) and mauve stingers (*Pelagia noctiluca*); while *T. dicentrarchi* has been identified in epidermal tissue from wild killer whales (*Orcinus orca*) and these animals may act as vectors for the bacteria [58,59,60] (Table A3). Recently, mortality events have also occurred in the kelp industry, Kombu (Saccharina japonica) seedlings experiencing green rotten disease had an increase in *Tenacibaculum* spp. from 0.8% to 4.5%; however, several other bacterial genera also experienced increases in abundance [61]. In conclusion, most *Tenacibaculum* sp. are not considered to be pathogenic and have few identified specific hosts; however, the eight pathogenic or potentially pathogenic *Tenacibaculum* sp. are often associated with fishes but have also been identified from a vast array of species that may act as vectors.

### 2.3. Putative Pathogens

#### 2.3.1. Identification of Pathogenic Species

In this section, the identification of potentially pathogenic species in the order of *T. ovolyticum*, *T. gallaicum*, *T. discolor*, *T. finnmarkense*, *T. mesophilum*, *T. soleae*, *T. dicentrarchi*, and *T. maritimum* is described. In addition, BLAST comparisons from the National Center for Biotechnology Information (NCBI; https://blast.ncbi.nlm.nih.gov/Blast.cgi) were used and the top 100 top hits described, with the search parameter “Organism” left blank unless otherwise mentioned.

##### *T. ovolyticum* 

Identification techniques *for T. ovolyticum* include DNA/RNA sequencing [41], MLSA [35,42], FAME (fatty acid methylesters) profile comparisons [62], and MALDI-TOF (matrix-assisted laser desorption/ionization time-of-flight mass spectrometry) [63]. The 16S rDNA sequence of *T. ovolyticum* IFO 15947 (Accession Number [AN]: NR_040912) is most similar to six isolates of *T. ovolyticum* (percent identity above 97.49%), the closest match to another species would be *T. dicentrarchi* AY7486TD (now *T. finnmarkense* AY7486TD [64]) and *T. soleae* LL04 12.1.7 with a percent identity of 96.88 and 96.54% (Appendix A). MLSA places *T. ovolyticum* in Clade two and its sister species is *T. soleae*, while *T. dicentrarchi* is the sister to both *T. ovolyticum* and *T. soleae* [35]. NCBI contains two complete genomes of *T. ovolyticum* (da5A-8, and DSM 18103) (Table A1) and based on GC content, size, and the number of coding sequences or genes, both complete genomes appear to be quite similar (Table A1). Non-genetic identification approaches applied to *T. ovolyticum* include FAME profile comparisons and MALDI-TOF. Two-dimensional plots comparing FAME profiles of *T. gallaicum*, *T. discolor*, *T. maritimum*, and *T. ovolyticum* indicated that *T. ovolyticum* was profoundly different, as fatty acids A35:0, Iso-A33:0, Iso-A34:0, A35:0 3OH, A35:1 ω6c, and anteiso-A35:0 had a greater mean percent composition, while Iso-A36:0 3OH had a lower mean percent composition [62]. The percent composition of numerous fatty acids was significantly different between the four species; however, more *Tenacibaculum* species need to be analyzed. FAME profiles can be used to help distinguish species identity but have also been correlated with pathogenicity in other bacterial species [65,66]. MALDI-TOF was applied for seven *Tenacibaculum* species and distinguished all tested species [63]. Several species-specific peak masses (2703.13, 5227.72, 9922.69, 10,239.19, and 10,582.04 *m*/*z*) unique to the *T. ovolyticum* isolate were identified [63]. Like FAME profiles, MALDI-TOF signatures have also been used to identify other bacterial species [67,68,69,70].

##### *T. gallaicum* and *T. discolor*

Identification techniques for *T. gallaicum* and *T. discolor* include 16S rDNA sequence comparisons, MLSA, MALDI-TOF, and FAME profiles. Representatives of these species (*T. gallaicum* A37.1T, *T. discolor* LL04 11.1.1T) were identified in the same study using morphology, GC content comparisons, DNA-DNA hybridization, and 16S rDNA sequences [12,13]. The 16S rDNA sequence of *T. gallaicum* BE263 (AN: LT601375.2) is most similar to three other *T. gallaicum* isolates (A37.1^T^, BE228, BE045), which had percent identities above 99.12% (Appendix A). Another comparison using *T. gallaicum* A37.1^T^ (AN: NR_042631.1) indicated several isolates of *T. litoreum*, *T. discolor*, *T. sediminilitoris*, and *T. ascidiaceicola* have high percent identities above 97.5% (Appendix A). 16S rDNA sequences of *T. discolor* 9A5 (AN: JQ231117.1), and LL04 11.1.1^T^ (AN: NR_042576.1) are most similar to five *T. discolor* isolates, *T. litoreum* CL-TF13, and *T. ascidiaceicola* RSS1-6, with percent identities above 99.10% (Appendix A). MLSA places *T. gallaicum* in Clade one, and the sister species was *T. litoreum*, while *T. discolor* was also found in Clade one, and was the sister to *T. mesophilum* [35]. Complete genomes of *T. gallaicum* DSM 18841, *T. discolor* DSM 18842 and IMLK18 are available online at NCBI (Table A1). FAME profiles for both *T. discolor* and *T. gallaicum* did not record species-specific differences [62]. There were also no specific masses through MALDI-TOF that allowed for distinction between *T. discolor* and *T. gallaicum*; however, the overall mass-spec could be used to identify either species [63]. The same study also noted that phyloproteomics indicated that isolates of *T. gallaicum* and *T. litoreum* had been misidentified and were proposed to be *T. discolor* [63].

##### *T. finnmarkense* 

*T. finnmarkense* has been identified using 16S rDNA sequences, MLSA and complete genome sequence comparisons. Comparisons using the 16S rDNA sequence of *T. finnmarkense* TNO006 (AN: MN699389.1) and S2F6 (AN: MF192947.1) demonstrated high sequence similarities among several *Tenacibaculum* species, similar to aforementioned comparisons using *T. gallaicum* and *T. discolor* (Appendix A). Other species had highly similar sequences, including *T. dicentrarchi*, *T. aestuarivivum*, *T. insulae*, *T. soleae*, and a *T. ovolyticum* clone (Appendix A). Genome (Illumina) sequencing and average nucleotide identity (ANI) of various *T. dicentrarchi* and *T. finnmarkense* isolates revealed that they were highly similar and were proposed to be sister-species [64]. However, it was also reported that *T. dicentrarchi* AYD7486TD was proposed to be within the species *T. finnmarkense* based on ANI [64] and MLSA [71]. Recently, MLSA and genome (Illumina) sequencing also indicated that the *T. finnmarkense* clade consists of two species (*T. finnmarkense* and *T. piscium*) and the *T. finnmarkense* species consists of two genomovars (*T. finnmarkense genomovar finnmarkense* and *T. finnmarkense genomovar ulcerans*) [72]. Complete genomes of *T. finnmarkense* (HFJ^T^ and TNO006), specific genomovars of *T. finnmarkense*, and *T. piscium* (TNO020^T^, TNO070, TNO063, TNO066, TNO064) are available on NCBI (Table A1) and more diagnostic techniques are required. FAME profiles of *T. piscium* (TNO020^T^), *T. finnmarkense genomovar ulcerans* (TNO010^T^), *T. finnmarkense genomovar finnmarkense* (TN0OO6^T^), *T. finnmarkense genomovar* (HFJ^T^), and *T. dicentrarchi* USC 35/09^T^ identified that numerous fatty acids were similar; however, the summed feature for *T. piscium* is greater than the other isolates, A35:0 is greater for *T. finnmarkense genomovar ulcerans* and *T. finnmarkense genomovar finnmarkense*, while *T. finnmarkense genomovar finnmarkense* had greater levels of A35:1 ω6c. More isolates need to be tested through FAME profiles to determine if any fatty acid could be used as a chemotaxonomic marker. MALDI-TOF [63] could be potentially useful for further describing clades of *T. finnmarkense* and closely related species.

##### *T. mesophilum* 

For the identification of *T. mesophilum*, 16S rDNA comparisons and MLSA have been applied. The 16S rDNA sequence of *T. mesophilum* MBIA3140 (AN: NR_024736) was most similar to 13 *T. mesophilum* isolates with percent identities above 99.04%; however, *T. lutimaris* DI 83II and *Actinobacterium* YH73 had percent identities of 97.44% and 99.29% (Appendix A). As mentioned previously, MLSA described a close relationship between *T. mesophilum* and *T. discolor* within Clade 1 [35]. Three full genomes of *T. mesophilum* are available on NCBI (Table A1). As for *T. finnmarkense*, more diagnostic tests are required, as well as studies to demonstrate the pathogenic potential of the bacteria.

##### *T. soleae* 

*T. soleae* has been identified using 16S rDNA sequences, PCR, MLSA, and MALDI-TOF. The 16S rDNA sequence of *T. soleae* LL04 12.1.7 (AN: NR_042630) indicated that all *T. soleae* isolates had percent identities above 98.15% (Appendix A). However, isolates of *T. dicentrarchi*, *T. aestuari*, *T. lutimaris*, *T. insulae*, and *T. discolor* had percent identities above 97.04% (Appendix A). BLAST comparisons indicate that PCR primer sequences using 16S rDNA [73] and 16S–23S internal spacer region (ISR) [74] were most similar to *T. soleae*; however, other outgroups had a percent identity of 100% for one of the primers. Both PCR assays have been applied before [75,76] and appears to be a reliable tool for detecting *T. soleae*. However, in another study, a new PCR assay specific to *T. soleae* was developed for multiplex-PCR due to previous PCR amplicons for other species of *Tenacibaculum* being a similar size [76]. The phantom band described in some samples [67] may be related to the amplicon of a potential contaminate, false positive, or another section within the genome that may amplify. For both PCR assays, melting curve analyses would help determine if the generation of multiple products is occurring. MLSA phylogenies placed *T. soleae* closest to *T. ovolyticum* [35]. The complete genome of *T. soleae* UCD-KL19 is available online (Table A1). The only non-genetic test, MALDI-TOF, applied to identify *T. soleae* was effective; a characteristic peak mass was identified at 9048.66 *m*/*z* [63]; meaning that a single peak could distinguish this species from the rest of the genus [63].

##### *T. dicentrarchi* 

*T. dicentrarchi* has been identified using 16S rDNA sequences, PCR, qPCR, MLSA, and MALDI-TOF. The 16S rDNA sequence of *T. dicentrarchi* 35/09^T^ (AN: NR_108475.1) indicated that isolates labeled as *T. dicentrarchi* had a percent identity above 98.21% (Appendix A). The most similar *Tenacibaculum* sequence of another named species would be *T. aestuariivivum* JDTF-79 and *T. ovolyticum* with a percent identity of 98.13% and 96.82% (Appendix A). A 16S rDNA PCR assay (Tenadi) to identify *T. dicentrarchi* [77] has limited application because it also identified *T. finnmarkense* [78]. A BLAST comparison of the Tenadi primers indicated that several outgroups including *T. finnmarkense* were matches (Appendix A). A new *T. dicentrarchi* PCR-specific assay was developed for multiplex-PCR due to the Tenadi assay amplicon being of a similar size to other species-specific PCR assays [76]; however, validation in other studies is needed. A 16S rDNA qPCR assay has been developed to identify *T. dicentrarchi* and was reported to be specific but needs to be validated and investigated for potential false positives [79]. MLSA demonstrated that *T. dicentrarchi* is in Clade two and was the phylogenetic sister to *T. ovolyticum* and *T. soleae* [35]. Subsequently, MLSA determined that the sister to *T. dicentrarchi* was an unnamed *Tenacibaculum* species (*T. finnmarkense*), while *T. soleae* and *T. ovolyticum* were still closely related in a monophyletic clade [35,42,64]. Four complete genomes of *T. dicentrarchi* are available on NCBI for comparison and include AY7486TD, TNO021, TD3509 = 35/09^T^, and TdChD05 (Table A1). Non-genetic identification techniques for *T. dicentrarchi* are limited to MALDI-TOF [63], which detected a peak mass unique to the species at 2579.41 *m*/*z* from *T. dicentrarchi* NCIMB14598 [63]. More species and isolates need to be tested to confirm if MALDI-TOF can distinguish between *T. finnmarkense* and *T. dicentrarchi*.

##### *T. maritimum* 

Many different techniques have been used for the identification of *T. maritimum* including rDNA sequence comparisons, PCR, nested PCR, qPCR, viability-qPCR (v-qPCR), MLSA, serology, FAME profiles, and MALDI-TOF. The 16S rDNA sequence of *T. maritimum* TmarCan1 (AN: KY428892.1) was most similar to ~30 *T. maritimum* isolates with a percent identity above 97.52% and the next closest comparison was *Polaribacter* sp. 7002-035 with a percent identity of 95.99%; other *Tenacibaculum* species have percent identities below 95.62% (Appendix A). Two 16S rDNA PCR assays [80,81] are widely applied to identify *T. maritimum* [82,83,84,85,86,87,88]. qPCR and v-qPCR for 16S rDNA [89,90,91], gyrB gene [92], and outer membrane protein A (ompA) gene sequences [7] have been developed. The 16S rDNA assay [89] was determined to be more sensitive than the ompA gene assay [7]. Using MLSA, *T. maritimum* was shown to be an independent lineage separated from monophyletic clades of other *Tenacibaculum* species, a classification that is also supported by the 16S rDNA alignment [35]. There are 25 complete genome sequences available on NCBI (Table A1). Most genetic identification techniques are specific to *T. maritimum*, with identification not being a primary concern for this species; novel studies may want to focus on the biology of *T. maritimum*.

Non-genetic identification techniques for *T. maritimum* include serology, FAME profiles, and MALDI-TOF. *T. maritimum* can be divided into at least four serogroups, which were identified among European or Asian isolates isolated from fishes [12,47,93,94,95]. Antigenic heterogeneity of *T. maritimum* was proposed to be host-specific [93]; however, at least two distinct serological groups of *T. maritimum* from Atlantic salmon in BC have been described [55,96]. Additional serological studies of isolates of *Tenacibaculum* spp. identified from mouthrot cases are needed to allow better characterization. Future studies should also focus on comparing isolates from the Americas against those from Europe and Asia to determine if the known serological scheme can be expanded. MALDI-TOF on 22 isolates for *T. maritimum* revealed that there was a species-specific peak mass at 9408.33 *m*/*z* and that there was also a characteristic peak mass at 11,356.67 *m*/*z* for 17 out of 22 isolates [63]. More recent MALDI-TOF applications identified 18 monomorphic and nine polymorphic biomarkers within the species, having potential use for species and strain typing [97]. The same study used MLST-like approaches combining isoform numbering (1–5) corresponding to the MALDI profile to designate a MALDI-type (MT1-20), and used the clustering of MALDI-types to identify MALDI-groups (MG1-4) [97]. Using the aforementioned technique, trends were identified between the geographical origin of the strain and the designated MT, in agreement with previous MLST results [35,97]. FAME profile could also distinguish *T. maritimum* from other species [62]; however, there were two main clusters (Ia and Ib) for *T. maritimum* and the author proposed that these clusters may be based on host species or geographic origin [62]. Based on the fatty acids iso-A35:1 G and iso-A35:0 3-OH comprising a greater mean percent composition for the tested *T. maritimum* isolates compared to the other species, it was also interpreted that those two fatty acids may be used as chemotaxonomic markers [62].

#### 2.3.2. Pathogenic Species

As mentioned above, eight *Tenacibaculum* species have been proposed as pathogens of marine finfish and possibly bivalves. Clinical signs of tenacibaculosis in marine fishes typically include external ulcers [6,7,9,12,16,26,98,99,100], frayed fins [12,14,98,100,101], pale organs [12,98,100,102], mortality [6,7,9,11,12,99], and atypical behaviors including lethargy [11,16,86,99], abnormal swimming (i.e., flashing) [16,26,74,100], and anorexia [16,26,99]. Three *Tenacibaculum* species have also been linked to disease in shellfish, including *T. maritimum*, *T. soleae*, and *T. mesophilum*.

##### *T. ovolyticum* 

*T. ovolyticum* has been found in deep waters off the coast of Japan [103], in a lobster culture associated with epizootic shell disease [104], as a component of sardine egg microflora [105] and has been reported to be an opportunistic pathogen in eggs and larvae of Atlantic halibut (*Hippoglossus hippoglossus* L.) [106,107]. In one study, when *T. ovolyticum* represented less than 30% of the epifloral community, halibut eggs hatched to larvae; however, when values rose above 30%, substantial increases in mortality at the hatching stage occurred [106]. *T. ovolyticum* was able to dissolve the chorion and damage the zona radiata through enzymatic activity [106]. Halibut but not turbot eggs immersed in baths of 10^5^–10^6^ bacteria mL^−1^ of *T. ovolyticum* before hatching had significantly increased mortality compared to controls [107]. In a phylogenetic study, of 89 *Tenacibaculum* isolates collected from disease outbreaks in Norway, isolate TNO089 had the greatest genetic similarity to *T. ovolyticum* (95%) and was cultured from halibut fry; demonstrating the potential for this organism to induce tenacibaculosis in fish past the larval stage [42]. Overall, *T. ovolyticum* has been found in environmental samples, invertebrates displaying disease and as part of the microflora of marine fish eggs; however, few studies to date have demonstrated that *T. ovolyticum* is a fish pathogen.

##### *T. gallaicum* and *T. discolor*

*T. gallaicum* and *T. discolor* were first identified together in Spain; *T. gallaicum* was isolated from seawater taken from a turbot (*Psetta maxima*) holding tank, and *T. discolor* was identified in the kidney of a deceased sole (*Solea senegalensis*) [12,13]. Both bacterial species are proposed pathogens because *T. discolor* L0LO4.11.1.1T and *T. gallaicum* A37.1^T^ experimentally induced tenacibaculosis in both turbot and sole [12,13]. For *T. discolor* and *T. gallaicum*, mortalities ranged 60–100% following intraperitoneal (IP) injections of 10^5^–10^7^ colony forming units (CFU) fish^−1^ [12,13]. Diseased sole and turbot displayed an eroded mouth, necrotic fins, ulcerations on the flanks, and pale internal organs [12,13]. Based on these findings, *T. gallaicum* and *T. discolor* are likely fish pathogens; however, more research is needed to validate a relationship between the presence of bacteria and diseased fishes and if these species are primary pathogens or opportunistic pathogens.

##### *T. finnmarkense* 

*T. finnmarkense* has been identified in Norway and Chile as a marine fish pathogen isolated from Atlantic salmon, lumpsuckers (*Cyclopterus lumpus* L.), coho salmon (*Oncorhynchus kisutch*), cleanerfish (*Symphodus melpops*), cod (*Gadus morhua*), and halibut (*Hippoglossus hippoglossus*) [11,40,64,71,72,86,101,108]. The bacteria have also been identified from the cnidarian, *Dipleurosoma typicum* [108]. The jellyfish was thought to be an unlikely vector for the bacteria, but its nematocysts can directly damage external salmon tissues facilitating infection [102]. Like *T. maritimum*, clinical signs of tenacibaculosis caused by *T. finnmarkense* included ulcerations on epidermal surfaces, frayed fins, and mortality [6,9,11,101]. Before the bacterium’s identity was confirmed to be *T. finnmarkense* in 2016 [40], several isolates (Tsp.1 and Tsp.2) were used in experimental trials. Atlantic salmon were exposed to 10^5^–10^8^ bacteria mL^−1^ for a set duration (1, 5, or 10 h) and were subsequently housed with naïve fish [11]. Fish exposed to Tsp.1 for 10 h experienced 100% mortality while all other groups had a mortality of roughly 30% and there was no mortality in the naïve cohabitants [11]. In another cohabitation trial, *S. salar* underwent bath immersions with 10^5^–10^6^ cells mL^−1^ of *T. finnmarkense* HFJ^T^ and Tsp.2 for 5 h, then exposed fish were grouped with naïve fish [101]. Isolate HFJ^T^ induced high mortalities (~80%) among infected fish; however, co-inhabitants (naïve fish) had fewer mortalities (<10%) [101]. Isolate Tsp.2 induced fewer mortalities in infected fish (<15%); however, naïve fish had similar mortalities (<20%) [101]. A notable difference between these studies was the stocking density used; 2.3 kg m^−3^ [11] compared to 4.6 kg m^−3^ [101]. Doubling the stocking density of shedders mixed with naïve fish would likely increase the chance of infected fish physically interacting with non-infected fish. The virulence of *T. finnmarkense* observed in *S. salar* may be isolate dependent. Determining the differences between isolates that induce disease in marine fishes and those that constitute a healthy microbiota require investigation [101].

##### *T. mesophilum* 

*T. mesophilum* was initially identified on a sponge (*Halichondria okadai*) [41], from sediment samples [109], and associated with the microbiome of Pacific white shrimp (*Litopenaeus vannamei*) [110]. There are few studies focused solely on *T. mesophilum*. Research has demonstrated that *T. mesophilum* induces a humoral immune response in gilthead seabream (*Sparus aurata*) [111] and produces a unique linear siderophore (bisucaberin (B) without macrocyclic counterparts [112,113]. Recently, the bacterium has been associated as the agent responsible for black-spot shell disease in Akoya pearl oysters (*Pinctada fucata*) [114]. A BLAST comparison (https://blast.ncbi.nlm.nih.gov/Blast.cgi) of 16S rDNA sequences of *Tenacibaculum* sp. Pbs-1 (NCBI Accession number: LC342074) cultured from diseased Akoya pearl oysters indicated that five separate sequences in the complete genome of *T. mesophilum* DSM 13764 (NCBI Accession number: CP045192) are identical (query cover of 100%, an E-value of 0, and a percent identity of 100%). Even though *Tenacibaculum* sp. strain Pbs-1 was thought to be the agent responsible for black-spot shell disease, reproduction of the disease including mortality requires additional factors, including a compromised shell [114]. More work is needed to demonstrate that *T. mesophilum* can be a pathogen to select groups of animals.

##### *T. soleae* 

*T. soleae* has been identified from marine environments around Europe and from fishes such as sole (*S. senegalensis*) [115], wedge sole (*Dicologoglossa cuneata* M.) [15], brill (*Scophthalmus rhombus* L.) [15], sea bass *(Dicentrarchus labrax*) [75], and wrasse [42]. *T. soleae* has also been reported in Pacific oysters (*Crassostrea gigas*) [116], as well as the American lobster (*Homarus americanus*) [104]. Bath infections of *T. soleae* using fry and juvenile wedge sole identified an LD_50_ of 7.8 × 10^5^ CFU mL^−1^ after an 18 h immersion and 100% mortality was observed in 6–8 days (d) using 10^7^ CFU mL^−1^ [15]. IP injections of the same isolate at a concentration of 10^6^ CFU fish^−1^ resulted with 100% mortality in five days [15]. Clinical signs in experimentally infected fish included external ulcers and erratic swimming behavior [15,75,115]. In *C. gigas*, clinical signs of disease included liquefactive necrosis in the adductor muscle and abnormal coloration of the mantle [116]. Experimental inoculation into *C. gigas* adductor muscles with 200 µL of 10^4^ CFU mL^−1^ of *T. soleae* resulted in cumulative mortalities of 46.6% and inoculations of 10^1^ CFU mL^−1^ induced mortality in one individual (6.7%) [116]. Further, the identification of *Tenacibaculum* sp. in abalone (*Haliotis laevigata* and *Haliotis discus hannai*) [117,118] and Akoya pearl oysters [114] suggest that tenacibaculosis is not specific to fish but has a much broader potential to infect other organisms such as bivalves.

##### *T. dicentrarchi* 

*T. dicentrarchi* is a potential pathogen that has received attention due to mortality events in Chile [14,98], and in BC, where the bacterium has been identified as a common isolate from *S. salar* with lesions similar to mouthrot [119]. In addition to Atlantic salmon [14], *T. dicentrarchi* has been documented in red conger eel (*Genypterus chilensis*) [98], and sea bass (*Dicentrarchus labrax*) [57]. Some *Tenacibaculum* sp. isolates from cod (*Gadus morhua*), wrasse species, and lumpfish (*Cyclopterus lumpus* L.) were also re-identified as *T. dicentrarchi* using MLSA [42]. Clinical signs in fishes infected with *T. dicentrarchi* included external ulcers, frayed fins, hemorrhagic organs, and damaged gills [14,57,98]. A bath immersion for 1 h using 3.78 × 10^5^ CFU mL^−1^ of *T. dicentrarchi* TdChD05 induced 65% mortality in *S. salar*, and 93% mortality in rainbow trout (*Oncorhynchus mykiss*); around 50% of the mortalities occurred in one day for *O. mykiss*, and four days for *S. salar* [14]. However, 50% cumulative mortality in *O. mykiss* observed within one day is unusually fast for the development of tenacibaculosis in fishes in comparison to other studies and may indicate that other factors were involved [6,7,14,15]. Coho salmon (*Oncorhynchus kisutch*) in the same experiment experienced no adverse effects [14], but outbreaks by *Tenacibaculum* spp. have been reported in coho salmon [71]. In a separate study, 30 Atlantic salmon smolts were IP injected with 10^7^ CFU fish^−1^ and then co-housed with another 197 salmon [120]. The water was then decreased from 1400 L to 400 L and the fish underwent bath immersion with 10^6^ CFU mL^−1^ for 30 min [120]. None of the fish in this trial died or displayed ulcers [120]. In a second trial, 26 of 36 fish were scarified using a scalpel blade to removes scales, and three drops of the bacterial culture were added to 20 of these scarified fish. All fish were then bath immersed in 10^7^ CFU fish^−1^ for 2 h. Ultimately, 32 fish died and 2 were euthanized as moribund [120]. Of the four survivors, one was scarified without the addition of bacteria and the three others only experienced the bath immersion [120]. All the fish were reported to have extensive scale loss, small hemorrhagic lesions, ascites, and dark livers [120]. Recently, exposure trials of Atlantic salmon to *Neoparamoeba perurans* (causative agent of amoebic gill disorder) identified a weak positive correlation between the presence of *N. perurans* and *T. dicentrarchi* in diseased fish [79]. In some lesions of diseased fish, 70.7% of the mean bacterial abundance was *T. dicentrarchi*, and there was reduced species richness and diversity indexes in diseased fish compared to naïve fish, supporting that dysbiosis may have implications for tenacibaculosis [79]. In another recent study, two isolates (QCR29 and QCR41) at 3.1 and 3.7 × 10^4^ CFU mL^−1^ were exposed to red conger eel (n = 12) for 2 h through bath exposure [121]. Eel mortalities began four days post-exposure and by the end of the experiment (30 d) 8 fish head died, and fish presented with epidermal ulcers, hemorrhagic fins, mouth and operculum, irritation around the head and yellow plaques around the jaws [121]. These experiments support the potential for *T. dicentrarchi* to be a pathogen of concern for aquaculture.

##### *T. maritimum* 

*T. maritimum* is the most documented *Tenacibaculum* species in the literature. The bacteria have been described worldwide, and in at least 30 host species (Table A3). In *S. salar* smolts, mortality rates in successful experimental infections often exceed 50% [6,7,9,26,122]. Variable success has been recorded using different infection protocols with the variability likely due to the isolate tested, host species used, infection methodology applied, and the concentration of infective dose [6,7,9,26,122].

#### 2.3.3. Pathogenesis

There is a limited amount of information available on the pathogenesis and virulence factors associated with the genus *Tenacibaculum* except for *T. maritimum*. The similarity of the clinical signs caused by several of the proposed pathogens of this genus suggest that common themes will apply as for *T. maritimum*. The pathogenesis section is described in the order of in-situ infections including the clinical presentation of mouthrot in BC, subcutaneous (SC) injections, IP injections, and bath immersion infections.

In-situ infections of *T. maritimum* are believed to infect fishes through damaged epithelia initially [26], where in Atlantic salmon, mouthrot is often associated with the mouth of fish [6,7]. In an *S. salar* smolt that died two months after being transferred to saltwater and was proposed to have died from mouthrot, histopathology and scanning electron microscopy (SEM) revealed lesions around the mouth surrounded by filamentous bacteria and dislodged teeth, with bacteria occupying gingival pockets [7]. Based on similar lesions from experimental bath infections using *T. maritimum* isolates (TmarCan15-1, TmarCan16-1, TmarCan16-2, and TmarCan16-5), it was proposed that for mouthrot, bacteria become systemic via the vascularized tooth pulp [7]. Histology from a jaw section of an Atlantic salmon with mouthrot had ulceration of the mucosal epithelium with plaques of basophilic filamentous bacteria and inflammation of the dermis, while bacteria were also dispersed throughout the compromised epithelium and dermis surrounding teeth (Figure 1). The histological section of the infected jaw presented here is similar to those of Frisch et al. [7]. While these authors also identified plaques of filamentous bacteria and necrosis of the gill tissue of affected Atlantic salmon. *T. maritimum* NCIMB 2153 in-situ infected farmed Senegalese sole (*S. senegalensis* L.) have also been studied by histology and SEM [47]. The flanks of infected fish exhibited a loss of epidermis, dermis, and hypodermis, extensive necrosis of superficial muscles, severe hemorrhages, and the presence of macrophages at infection sites [47]. SEM demonstrated that some lesions first impacted the epithelium in the middle of a scale exposing the fibroid bone and then progressed outwards. SEM also revealed that areas without epithelium had copious amounts of rod-shaped bacteria present around scales and a reduction in the micro-ridges of the surface epithelia [47]. Further studies are required as there are substantial differences between the reports from [7,47] including fish species and bacterial isolates used, different geographic locations, and different transmission routes, resulting in different clinical signs of infection. From Atlantic salmon post-smolts sampled at BC netpen sites, microbial profiling using the 16S rDNA sequences indicated that *T. maritimum* was identified in healthy, diseased, and post-treated fish [123]. Diseased fish had reduced microbial diversities with two sequence variants of *T. maritimum* dominating the community [123]. Surviving fish with high proportions of *T. maritimum*, in association with the abundance *Vibrio* spp. and the presence of mouthrot led researchers to provide evidence that mouthrot is a complex multifactorial disease characterized by dysbiosis [123]. With several *Tenacibaculum* species reported to have associations with dysbiosis, more research should focus on what multifactorial processes initiate dysbiosis linked to the presence of mouthrot. In BC, the most common clinical presentation of tenacibaculosis is denoted as mouthrot. The few common clinical signs of mouthrot include yellow plaques and ulcers on the mandibles, gills, and infrequently on the flank (Figure 2). The distinguishing feature of mouthrot compared to other presentations of tenacibaculosis is that at netpen sites, small plaques often occur on the jaws and gills with few conspicuous ulcers (Figure 2). Ulcers can be found on other epidermal surfaces, and laboratory trials identify more severe ulcerations on the fish [6,7,8]. In other presentations of tenacibaculosis, conspicuous ulcers are often located around the head, flanks, and fins of the fish [26,49,99,100].

SC and IP infection trials using *T. maritimum* has been documented in several fishes. In turbot, SC and IP injections of 10^8^–10^9^ CFU fish^−1^ of *T. maritimum* LL01.8.3.8 induced anorexia and lethargy as early as 3 h post-injection (hpi) [99]. At this time, histopathology of tissue at the site of SC injection demonstrated clusters of bacteria distributed through the connective tissue of the hypodermis and degeneration of muscle without inflammatory responses [99]. The first gross lesion was recorded at 24 hpi and consisted of discoloration at the injection site [99]. The formation of an ulcer at the inoculation site occurred around 48 hpi, while at 72 hpi, ulcers obtained the characteristic circular appearance and developed peripheral hyperemia [99]. At all-time points, histopathology demonstrated degeneration and necrosis of muscles, detached or absent epidermis and dermis, and inflammation of the ulcerated area [99]. The expansion of the lesion was evident based on the spread of grossly discolored tissues [99]. At seven days post-infection, most fish displayed ulcerative dermatitis and hyperemia and some fish displayed diffuse hyperemia and hemorrhages on the fins [99]. Throughout the experiment, bacteria were primarily found from three organs (skin, kidney, and spleen) using PCR, culture, and immunohistochemistry [99]. A subsequent study used SC injections of 10^8^ CFU fish^−1^
*T. maritimum* LL01.8.3.8 on turbot, and similar pathological signs were noted [10]. In another study, Dover sole (*Solea solea* L.) injected subdermally with 10^7^ cells fish^−1^
*T. maritimum* developed epidermal lesions and experienced 100% mortality in four days [125]. Difficulties were also reported based on bacterial cultures flocculating leading to inconsistent quantification of bacteria and numerous mortalities in both pilot trials [125]. In the first pilot trial, SC-injected fish had 30% mortality in 48 h and the rest were euthanized; in the second pilot trial, the proportion of mortalities among controls were 30% and 50% [125]. The high number of mortalities within such short time frames for both pilot and experimental trials may indicate that other factors may be involved. In black sea bream (*Acanthopagrus schlegei*), SC injection of 10^2^–10^6^ cells fish^−1^ using *T. maritimum* A4 induced ulcers at the injection site [126]. However, since the highest mortality using 10^6^ cells fish^−1^ was 52.6% while the control group experienced 40.8%, it is possible that the infection method may have contributed to mortality [126]. Based on these results, the experimental reliability and effectiveness of SC injections are questionable and additional studies are needed to validate current work in these fish species.

IP injections of 10^8^–10^9^ CFU fish^−1^ of *T. maritimum* strain LL01.8.3.8 in turbot at 168 hpi resulted in splenitis and capsular necrosis, necrosis and hemorrhage of the liver, head kidney, and intestine, reduced hematopoietic tissue in the head and kidney, and enteritis with transmural inflammation [99]. No grossly observable, macroscopic epidermal lesions were recorded, but *T. maritimum* was identified in the liver, heart, gastrointestinal tract, and gills from 6–48 hpi and the spleen and kidney from 6–168 hpi using immunohistochemistry [99]. In Atlantic salmon, isolated extracellular proteins (>250 µg/fish) from *T. maritimum* 89/4762 injected-IP induced 100% mortality, with the author concluding that extracellular products released from *T. maritimum* induce disease [122]. Overall, it appears that IP injections of *T. maritimum* can produce internal lesions, and extracellular products may play a significant role in the pathogenesis of disease. However, other studies were unsuccessful in reproducing tenacibaculosis through IP injections [9,127].

For both IP and SC infection methodologies, bacteria were detected in most organs as early as 3 h until the end of the experiment; bacteria were also detected in blood vessels, possibly demonstrating bacteremia [99]. However, IP and SC injections do not mimic natural horizontal transmission, and researchers have had variable success inducing infection [6,7,9,127]. Subcutaneously injected fish displayed similar clinical signs to affected fish from production settings [99].

Bath infections, mimicking horizontal transmission, using *T. maritimum*, also resulted in clinical signs similar to those described from outbreaks on fish farms [6]. Infection trials with turbot using *T. maritimum* isolate ACC6.1 suggested that an immersion of 18 h using 5 × 10^3^ CFU mL^−1^ was needed [128]. However, in Atlantic salmon, successful bath infections have occurred at 1.5 and 5 h using various Canadian *T. maritimum* isolates at 10^7^ CFU mL^−1^ [6]. Isolate virulence and bath concentration are likely important factors that impact the period required to allow infection [6]. Bath infections with *S. salar* smolts produced atypical behavior (erratic swimming and loss of equilibrium), oral ulcers, and yellow plaques on the external surfaces such as the mouth and gill [6,7]. In several fishes including *S. salar*, bath infections resulted in ulcers externally on the flanks and fins of fishes, pale organs, friable livers, congested kidneys, and eventually mortality [16,26,100,102,129]. Numerous *Tenacibaculum* sp. infections in laboratory practices use variably concentrated infective doses, often at high concentrations (above 10^3^ CFU mL^−1^), which may indicate that the bacteria are not primary pathogens, but instead other variables, including dysbiosis [123], may influence infection.

#### 2.3.4. Virulence Factors

Virulence factors are required to allow bacteria to invade, induce disease, and evade host defenses [130]. A complete genome analysis of *T. maritimum* NCIMB 2154^T^ identified categories of virulence genes related to motility, adhesion, quorum sensing/quenching, metabolism, iron acquisition, stress response, transport/secretion systems, and toxins [44].

*T. maritimum* genes predicted to express proteins for gliding machinery (14 gld genes [gldA to gldN] and 10 spr genes [sprA to sprE, and five sprF paralogs]) allowing mobility on multiple surfaces have been identified. Seventeen other genes in *T. maritimum* code for various adhesins, factors related to the biosynthesis of exopolysaccharides, and lectin or carbohydrate-binding motifs [44]. These three groups (adhesins, exopolysaccharides, and binding motifs) may allow adhesion to multiple biotic and abiotic surfaces [44]. Several isolates of *T. dicentrarchi* and *T. maritimum* from *S. salar* lesions formed biofilms on abiotic surfaces such as polystyrene [131,132]. During the same experiment, biofilm formation indexes were the greatest at 24 h for all strains tested, but there was significant variability between strains over time (120 h) [131,132]. More studies are needed to determine what adhesins are necessary to bind to specific surfaces, as one researcher has highlighted the difficulty in creating infection models for fishes based on a lack of understanding of the adhesive properties of *Tenacibaculum* species [6]. An understanding of the adhesive properties of *T. maritimum* may help identify the mechanisms that lead to flocculation in media, as this can lead to unreliable estimates in bacterial concentration using spectrophotometry and has downstream implications for experimental trials.

Quorum sensing and quenching have been reported in *T. maritimum*. *N*-acyl homoserine lactone (AHL) activity was identified among nine strains of *T. maritimum*; strain NCIMB 2154^T^ possessed *N*-butyryl-l-homoserine lactone (C4-HSL) and was also capable of degrading long-acyl AHLs (A30-HSL) [46]. No gene for AHL biosynthesis was identified, and no genes for quorum quenching were identified in strain NCIMB 2154^T^, but an AHL lactonase encoding gene was predicted based on its identification in *T. discolor* strain 20 J [44,133]. In another study, multiple AHLs (C6-HSL, 3-oxo-C6-HSL, C8-HSL, 3-oxo-C8-HSL, A30-HSL, 3-oxo-A30-HSL, A32-HSL, 3-oxo-A32-HSL, A34-HSL, and 3-oxo-A34-HSL) were identified in *T. discolor* and *T. soleae* but a lactonase enzyme was only identified in *T. soleae* [134]. Quorum quenching enzymes such as lactonase may be different at the species and strain level in bacteria [134], possibly because, depending on the bacterial community, different compounds may be more effective for communication.

Genes in *T. maritimum* related to metabolism include a complete glycolysis pathway, a tricarboxylic acid cycle, sugar transporters, sugar enzymes, and several proteases, among others [44]. The identification of various proteases, which degrade proteinaceous compounds such as gelatin and casein, and the capability to use amino acids as a carbon and nitrogen source, support this organism’s capability as a pathogen and its ability to survive off the host [44,50]. However, the identification of genes related to carbohydrate processing is a unique finding and contradicts previous studies that demonstrated that the bacterium is unable to process simple and complex carbohydrates [9,44,50]. Determining the function of these genes may explain how nutrients are obtained and utilized by *Tenacibaculum*.

Iron acquisition genes have been identified in *T. maritimum* and include the production of the bisucaberin siderophores and transporters, heme-related proteins, iron-regulation proteins, as well as a Fur regulator [44]. The identification of these genes is in agreement with the results obtained from iron-limitation experiments and assays [135]. *T. maritimum* isolates utilized several iron sources (hemin, hemoglobin, ferric ammonic citrate, and transferrin) when added to iron-deficient media, were able to bind to hemin, demonstrating the presence of heme-related proteins, and had siderophores identified in universal colorimetric chemical assays [135]. Hypothetical genes involved in iron acquisition [44] and the ability to remove iron from other sources [135] may play important roles in obtaining iron from the blood and tissue, as *T. maritimum* are reported to undergo bacteremia and can produce lesions in the liver and spleen [118,128,129]. Future studies should investigate how these genes aid iron regulation in *Tenacibaculum* species and should identify the expression of these genes in in-vivo/vitro models.

Bacterial stressors can include chemical (i.e., reactive oxygen species (ROS), heavy metals), physical (i.e., temperature), and biological interactions. *T. maritimum* encodes three superoxide dismutases (SodA, SodB, and SodC) and two catalase/peroxidase enzymes (KatA and KatG), indicating that the organism can cope with oxidative stress [44]. Applications of hydrogen peroxide, which generate ROS, did not dramatically reduce *T. maritimum* infections but speculated that hydrogen peroxide inadvertently promoted tenacibaculosis through the stress that the fish experienced during treatment [136]. Several genes related to heavy metal resistance have also been identified and were proposed to remove cationic heavy metals to limit ROS production [44]. Temperature is another stressor, but there is considerable variation in the range of temperatures tolerated by *Tenacibaculum* species (Table A2). Studies have reported fish mortalities caused by *Tenacibaculum* spp. following either a decrease or increase in water temperature [26,103]. Additional research should occur to identify genes related to stressor response.

Transport systems are useful for pathogens, as they allow proteins to be brought to the cell surface. Genes encoding an ATP binding cassette type transport system, a Sec-dependent transport system, a twin-arginine transport system, and a type IX secretion system (T9SS) were identified in *T. maritimum* [44]. Extracellular products of *T. maritimum* reported to induce mortality in *S. salar* by [117] are possibly transported using these systems. The role of each transport system in *Tenacibaculum* sp. needs further research; if toxins can be prevented from reaching the surface of the bacterial cell, *Tenacibaculum* infections may be attenuated.

*T. maritimum* genes have been found that code for many enzymes, including cholesterol-dependent cytolysin, collagenase, sphingomyelinase, ceramidase, chondroitin AC lyase, streptopain family protease, and proteins related to sialoglycan degradation/uptake [44,97]. Many of these enzymes are classified as toxins because they damage cells. For example, cholesterol-dependent cytolysins are cytolytic pore-forming toxins; however, these are also predicted to interact with the phagosome (as with *Listeria* monocytogenes) or cause translocation of enzymes (as with *Streptococcus* pyogenes) [137]. Sphingomyelinases are multi-functional and can aid phagosomal escape or avoidance, tissue colonization, infection establishment, and evasion from host immune responses [138]. Ceramidase in *Pseudomonas aeruginosa* has been reported to have functions linked to hemolysis in mammals [139]. Given that heme-related genes were identified in *T. maritimum*, ceramidase may be linked to iron acquisition if ceramidase can lyse cells rich in iron, such as erythrocytes in the bloodstream of fishes. Chondroitin AC lyase hydrolyzes chondroitin; a reduction in the rigidity of connective tissues caused by loss of chondroitin allows for easier dissemination of bacteria throughout the host [140]. Collagenase is an enzyme that breaks down collagen [141]. Since the skin, cartilage, and bones of finfish are rich in chondroitin sulphate and collagen [141,142,143], chondroitin AC lyase and collagenase may play a role in the development of external lesions and invasion into deeper tissues. Genes for sialidase were reported, and their products may allow foraging for host glycoproteins [44,144]. Further research is needed to determine how the genes identified by [44] are related to the pathogenesis of disease induced by *T. maritimum.*

## 3. Conclusions

Members of the genus *Tenacibaculum* are Gram-negative, filamentous, marine bacteria that are likely cosmopolitan and ubiquitous. Most bacterial species are non-pathogenic or have not been reported in mortality events, while eight other species (*T. ovolyticum*, *T. gallaicum*, *T. discolor*, *T. finnmarkense*, *T. mesophilum*, *T. soleae*, *T. dicentrarchi*, and *T. maritimum*) have been related to finfish or shellfish mortality events. Most potential pathogens are identified using 16S rDNA sequencing, and few diagnostic tests have been developed to identify each species, except *T. maritimum*. Similar clinical signs of infection in fishes induced by *Tenacibaculum* spp. include external ulcers, atypical behaviors, and mortality, and indicate that the term tenacibaculosis should be expanded to encompass *Tenacibaculum* species. Imitations of tenacibaculosis outbreaks from aquaculture sites are repeatable using experimental infections via bath immersions, where SC and IP injections have had less success. Variations in observed infections can be related to the bacterial isolate, host, geographic origin, and mode of transmission. More research is needed to define local distributions of bacteria, increase the number of diagnostic tests for pathogenic species, and clarify the pathogenesis of *Tenacibaculum* species.

## Figures and Tables

**Figure 1 pathogens-09-01029-f001:**
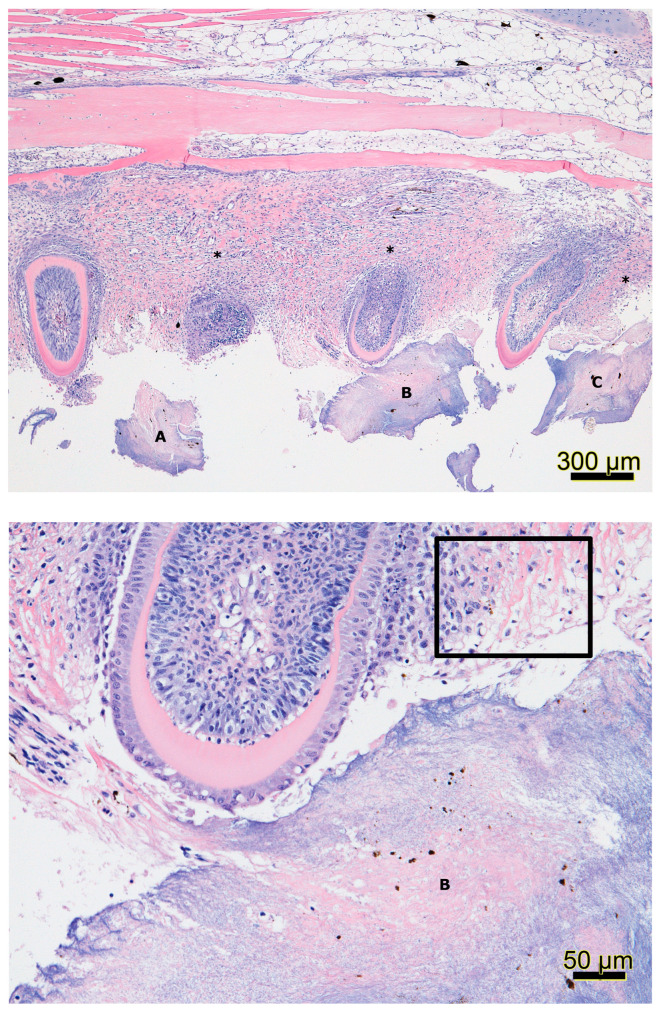
Histology of an Atlantic salmon (*Salmo salar*) with mouthrot in British Columbia (Canada). Top: There is extensive ulceration of the mucosal epithelium with thick basophilic plaques (**A**–**C**) and marked diffuse inflammation of the dermis (*). Bottom: A magnified portion of a tooth and a plaque of filamentous bacteria (**B**), where bacteria are seen disseminating into surrounding tissues with surrounding macrophages, necrotic cells, and fibrin and with disruption of connective tissue (black square). Optimization of photomicrograph illumination and color balance followed published methods [124] and was provided by G. M. Marty.

**Figure 2 pathogens-09-01029-f002:**
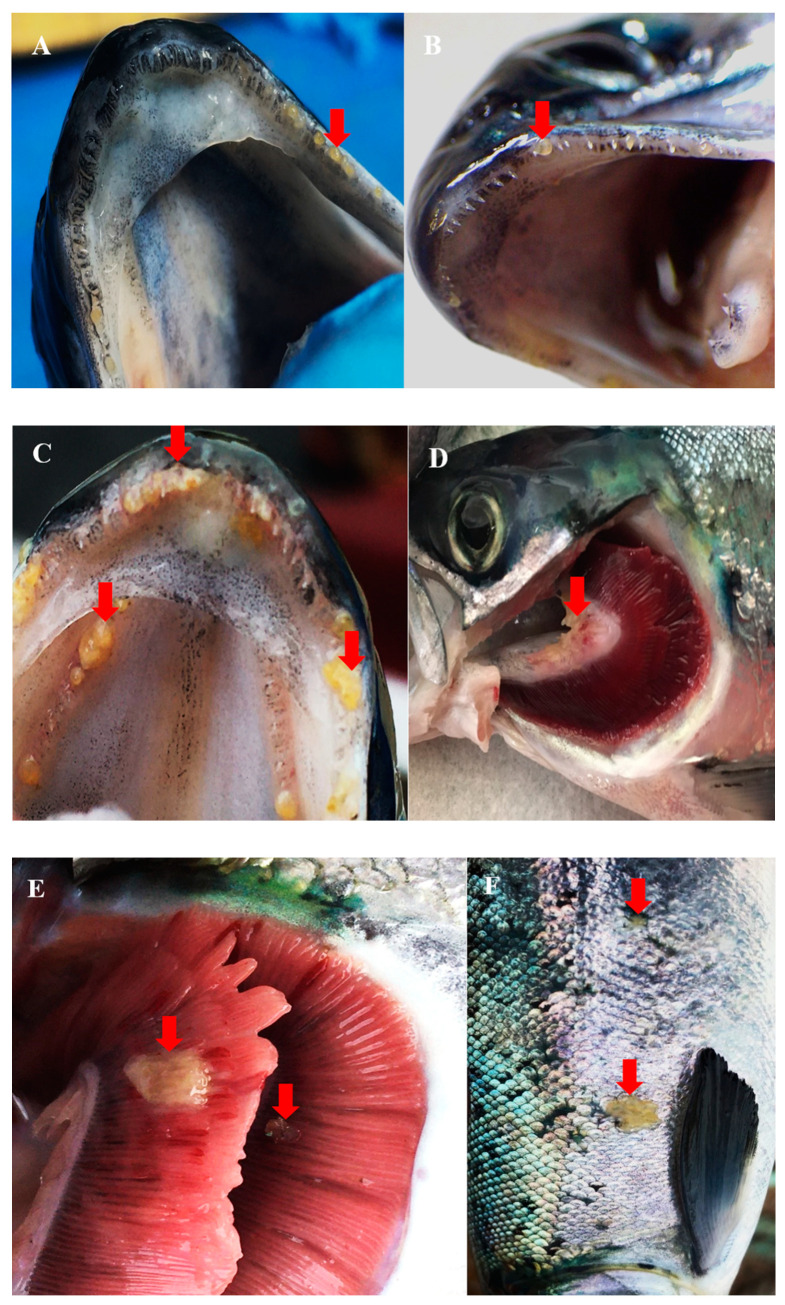
Clinical presentation of mouthrot in Atlantic salmon (*Salmo salar*) in British Columbia (Canada). (**A**–**C**): Yellow plaques (arrow) on the pre-maxilla, maxilla, and palatine teeth. (**D**,**E**) Yellow plaques (arrow) on the gill-raker or the gill filaments, while discoloration, clumping, and hemorrhaging of the filaments is also present. F: Two yellow plaques (arrow) on the flank on the fish. Pictures are courtesy of the Fish Health Team of Mowi Canada West.

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
