# Peer review of "Advancements in Characterizing Tenacibaculum Infections in Canada"

_pathogens, 2020, doi:10.3390/pathogens9121029_

Round 1

Reviewer 1 Report

The review article discusses the bacterial pathogen Tenacibaculum species diversity and the disease tenacibaculosis. In the aquaculture industry, Tenacibaculum infections are poorly documented. This is unique reviewed article focuses on major pathogenic Tenacibaculum species. This article reviewed the ecology, identification of pathogenic species distributed in the Canadian aquaculture industry. This type of review article is highly necessary for the discovery of new emerging aquatic pathogens.

This review manuscript combines most of the published research articles on pathogenic Tenacibaculum species. One can get a clear picture of the current status of this pathogen in the aquaculture industry by reading the article.

I would like to appreciate the authors for their extensive literature search and well-organized discussion. More information or background research can be added to the “introduction” section. 

Please format the tables properly in the Appendix-B1

Author Response

Please see the attachment, the first page contains the reviewer's feedback, the second contains the responses to their feedback.

Reviewer 2 Report

Review

Literature review: Advancements in characterizing Tenacibaculum infections in Canada

The authors present a Literature review about the characterisation of potentially pathogenic members of the genus Tenacibaculum, their ecology, identification and impact with focus on Atlantic salmon in Canada. The background is that the impact of mouthrot seems to be increasing in British Columbia. They point at advancements made in the characterizing of Tenacibaculum sp. infections and identify knowledge gaps.

The review is well written and topics well organized and to this reviewer the authors seem to have been able to include most of the literature available within their scope. This overview is relevant and needed at this stage.

Details

Line 31, references 4 and 5: Please, provide web links, if possible.

Lines 31 and 37: To this reviewers understanding, the manifestation of tenacibaculosis in BC, ‘mouthrot’, is seen as yellow plaques within the oral cavity, and is atypical and different from classical tenacibaculosis involving skin lesions, mouth erosions (external), frayed fins and tail rot. This should be clear from the beginning of the paper. Preferably the mouthrot here should be described and accompanied with references available to the global reader (4 and 5 are not).

Line 33: Include reference 97? Please, provide a link to reference 10, as it could not be found.

Paragraph lines 52-65: This information is interesting, but ought to be less anecdotal. References? 

Lines 62-63: This sentence is unclear. Please, rewrite.

Line 68: Reference 18 is not reachable. Please, provide a link.

Line 93: Please, use www.bacterio.net/flavobacteriaceae.html

Line 134: As RTFS usually have a very different manifestation than tenacibaculosis, this sentence should be slightly rewritten.

Line 153: Multilocus (one word)

Lines 157 and 158: Correct spellings: T. geojense, T. skagerrakense, T. litopeneai

Lines 171-172: This reviewer found this information in paper 42: ‘ chromosome with predicted 2866 protein coding genes’. Please, check.

Line 187: Include paper 97 for the use of marine agar. Flexibacter maritimus medium (not media). Cytophaga agar prepared in sea water could also be included (paper 134).

Line 209: T. maritimum is also reported from Norway, see papers 83 and 40.

Lines 223 and 224: Reference 10 is not available to this reviewer. The author is also co-author of paper 99, where they state that T. finnmarkense, as yet, is not shown as the causative agent, until their presentation in paper 99. The first paper showing T. finnmarkense (T. sp. group 1) as causative of the typical skin and cornea ulcers, however, seems to be paper 97?

Paragraph 242 - : Should paper 104 be included?

Line 247: This reviewer would already at this point appreciate to become aware of the proposal that the T. dicentrarchi AY7486TD strain rather belongs to the species T. finnmarkense (ref. 67).

Line 268: Why is paper 11 included here?

Line 299: This reviewer has already become aware of FAME profiles reported in reference 69.

Line 359: Paper 33 should be included here.

Line 359: Are references 30 and 37 relevant?

Lines 420-21: In paper 69 T. finnmarkense is reported also from cleanerfish (Symphodus melpops), cod (Gadus morhua) and halibut (Hippoglossus hippoglossus).

Line 460: In paper 40 T. soleae is reported from wrasse.

Line 478: To this reviewers understanding of paper 40, the MLSA placed Tenacibaculum sp. isolates and not T. maritimum isolates as T. dicentrarchi. Please, correct.

Line 522: Paper 24 is stating that there is a consistency in pathology in salmonids and non-salmonids with erosive lesions of external surfaces being the most prominent clinical sign. Head lesions are also observed, but to this reviewers understanding, paper 24 does not support that the mouth is “often” involved, and in case of head lesions, the findings are external. As the following details in the paragraph is describing a condition in BC “different from classical tenacibaculuosis” (8), occurring inside the mouth, the atypical, but nevertheless interesting, character, of this manifestation should be clarified to the reader.

Lines 571-626: Include a comment on the magnitude of infection doses used in the transmission trials.

Lines 729-730: Include water temperature?

Appendix A: The list seems only partly updated according to the very recent paper 69.

Lines 762-763: In paper 148 T. ascidiaceicola is reported with no anaerobic growth. Please, correct.

Appendix B Table: T. finnmarkense TNO006 and TNO010 are from Norway (67, 69).

Author Response

Please see the attachment, the first and second page has the reviewer's feedback, the other pages have the responses to the reviewer.

Reviewer 3 Report

The argument is well faced up but it seems more oriented to fish than the bivalves. For this purpose, please give more argument to shellfish in order to give equal attention to both the categories of aquatic organisms. Also, for the same purpose, in the Abstract, add a sentence regarding shellfish as reported for clinical signs for fish.

Author Response

Please see the attachment, the first page has the reviewer's feedback, the second page has the responses to the reviewer.
